# Age-dependent virulence of human pathogens

**Gabriele Sorci** [ID]*, **Bruno Faivre**

Biogéosciences, UMR 6282 CNRS, Université de Bourgogne Franche-Comté, Dijon, France

* gabriele.sorci@u-bourgogne.fr

**Data Availability Statement:** All relevant data are within the manuscript and its Supporting Information files.

**Funding:** GS acknowledges support under grant number ANR-21-CE35-0015 from the Agence Nationale de la Recherche, https://anr.fr/. The

## Abstract

Host age is often evoked as an intrinsic factor aggravating the outcome of host-pathogen interactions. However, the shape of the relationship between age and infection-induced mortality might differ among pathogens, with specific clinical and ecological traits making some pathogens more likely to exert higher mortality in older hosts. Here, we used a large dataset on age-specific case fatality rate (CFR) of 28 human infectious diseases to investigate i) whether age is consistently associated to increased CFR, ii) whether pathogen characteristics might explain higher CFR in older adults. We found that, for most of the infectious diseases considered here, CFR slightly decreased during the first years of life and then steeply increased in older adults. Pathogens inducing diseases with long-lasting symptoms had the steepest increase of age-dependent CFR. Similarly, bacterial diseases and emerging viruses were associated with increasing mortality risk in the oldest age classes. On the contrary, we did not find evidence suggesting that systemic infections have steeper slopes between CFR and age; similarly, the relationship between age and CFR did not differ according to the pathogen transmission mode. Overall, our analysis shows that age is a key trait affecting infection-induced mortality rate in humans, and that the extent of the aggravating effect on older adults depends on some key traits, such as the duration of illness.

## Author summary

Mortality due to infectious diseases varies tremendously among infectious agents, with some pathogens producing no mortality, and others being often associated with a fatal outcome. Such variability depends on characteristics of the pathogen, the host and the environment where hosts and pathogens interact. Age is one of the main host traits that accounts for differences in infection-induced mortality (with mortality being higher at the extremes of the age spectrum). Here, we used a large dataset on 28 human infectious diseases to explore the clinical and ecological traits that might account for differences in age-specific mortality risk. We found that pathogens producing long-lasting disease symptoms exert the highest mortality risk in the older adults. Similarly, emerging pathogens are also associated with higher mortality risk in the oldest age classes. These results confirm that age is a key trait affecting infection-induced mortality rate in humans, and show that the

funder had no role in study design data collection, and analysis, decision to publish or preparation of the manuscript.

**Competing interests:** The authors have declared that no competing interests exist.

extent of the aggravating effect in older adults depends on some key traits, such as the duration of illness.

## Introduction

Throughout their history, humans have paid a high toll to infectious diseases [1,2]. Although the improvement of hygiene, the discovery of antibiotics, and the implementation of large-scale vaccination programs have led to a reduction in infection-induced mortality (e.g., [3]), the hope that we might win the war against pathogenic microorganisms has been dashed by the emergence of novel diseases from animal reservoirs and the evolution of multi-drug resistant strains [4,5].

Pathogen virulence is often conceived as the damage caused by the infection, potentially leading to host mortality [6]. Virulence is a complex trait that depends on features of the pathogen, the host and the environment where the interaction takes place [7]. Case fatality rate (CFR) is a metric defining the severity of human infectious diseases [8,9], and refers to the probability of dying after contracting the infection, in the absence of pharmaceutical treatments (untreated CFR), or when treatments are available (treated CFR). CFR varies tremendously among human infection diseases [8,10–11]. However, as mentioned above, the sources of this variability can be found in pathogen, host, and environmental traits.

The SARS-CoV-2 pandemic provides an excellent illustration of the idea that pathogen virulence does depend on the combination of pathogen, host and environmental traits. Indeed, some viral strains have been shown to induce more severe symptoms than others (e.g., delta vs. omicron variants [12]), hosts with comorbidities have a higher probability to succumb to the infection [13,14] and there exists extensive geographic (among-country) variation in SARS-CoV-2 induced mortality [15].

Age is one of the host traits likely to have profound consequences for pathogen virulence [16]. Again, following the emergence of SARS-CoV-2, it was rapidly established that CFR dramatically differed among age classes, with infection-induced mortality being essentially nil in children and substantial among the oldest age classes [17,18]. The recognition of host age as an important determinant of pathogen virulence is not new. Changes in infection-induced mortality across ages have been historically reported for several infectious diseases (see [19,20] and references therein). Actually, this old literature often reported non-linear relationships between age and severity of infection, with children in pre-puberal age having the lowest risk to succumb to the infection (the so-called honeymoon period of infectious diseases [21]). Therefore, the relationship between age and infection-induced mortality is usually thought to be J-shaped [19,20,22].

The reasons underlying this non-linear relationship between infection-induced mortality and age are not fully understood, but it is generally assumed that age-related changes in immune function are involved [21]. The immune system goes through a series of developmental stages during the ontogeny, with further changes occurring during aging [23–25]. A full description of age-dependent changes in immune function is well beyond the scope of this article. Nevertheless, the immune response at the extremes of the age spectrum can be summarized by weak innate immunity (e.g., impaired neutrophil functions, reduced TLR4 expression and cytokine production) and highly regulated immune responses (an anti-inflammatory profile) in newborns; and depleted naïve T cells, impaired innate immunity and low-grade pro-inflammatory profile in the oldest adults [23–25]. Accordingly, higher susceptibility to infectious diseases at the extremes of the age spectrum might reflect an immature (or a hyper-

regulated [26]) immune system in infants and a senescing one in older adults [25]. This hypothesis rests on the assumption that infection-induced mortality is essentially due to impaired resistance, defined as the capacity to limit pathogen multiplication [27]. Under this assumption, if the immune system of infants and older adults is indeed less competent to fight infections, mortality burden is expected to be the highest in these age classes. However, there is compelling evidence showing that infection-induced mortality is not merely a matter of over-whelming pathogen proliferation due to failure to produce an effective immune response [28,29]. Actually, over reacting immune responses are often the most important causes of infection-induced mortality [28]. Therefore, age-dependent mortality might rather reflect differences in the proper regulation of the inflammatory response, and thus avoid immunopathology damage in old individuals. Finally, increased vulnerability to infectious diseases at the extremes of the age spectrum might reflect differences in the capacity to repair tissues and organs that have been damaged by the infection (and/or the inflammatory response produced following the infection) [30–32]. This last point, therefore, suggests that tolerance (i.e., the capacity to minimize the infection-induced damage) might also differ across ages and account for differences in age-specific CFRs [33].

As mentioned above, focusing on host traits (e.g., host immunity) to explain variation in CFR might provide an incomplete picture since pathogen traits can also determine whether some age classes are more prone to succumb to the infection than others. After they enter the host, pathogens differ in their organ tropism, and can induce localized or systemic (multi-organ) infections. Similarly, pathogens differ in the time needed to produce disease symptoms (i.e., the incubation period) and in their duration. For instance, while campylobacteriosis (a food-borne disease caused by bacteria of the genus *Campylobacter*) produces symptoms typically lasting 3 to 6 days [34], illness caused by hepatitis A usually lasts for months, with 10% to 15% of patients having prolonged symptoms for up to 6 months [35]. Human pathogens can also differ in the degree of host adaptation, which can contribute to determine the severity of the infection. By definition, pathogens that only infect humans should be more human-adapted than pathogens that are only transmitted from animal reservoirs, and it has been suggested that this could affect virulence [11]. Finally, pathogens can be transmitted from host to host through different pathways, such as food-borne, air-borne or vector-borne diseases, and previous work has shown that transmission mode can be associated with pathogen virulence [6,11,36].

While the association between pathogen traits and virulence has already been explored, to the best of our knowledge, how pathogen features shape age-dependent virulence has not been investigated, yet. Here, we try to fill this gap. To this end, we used a large dataset on age-specific infection-induced mortality that has been recently compiled [20]. Glynn and Moss [20] showed that the severity of most infectious diseases was lowest for school-age children. We went a step further and for each infectious disease included in the dataset, we added the following information: 1) whether the infectious agent is a virus or a bacterium; 2) the duration of the incubation period; 3) the duration of the disease symptoms; 4) whether the infection is localized or systemic; 5) whether the infectious agent can be transmitted from human-to-human; 6) whether the infectious agent has an animal reservoir; 8) whether the infection agent has a recent history of human infection (i.e., has emerged in 20th century); 7) whether the infectious agent is transmitted by ingestion, inhalation, contact with body fluids, or by vectors.

Based on the rationale that age might reduce both host resistance and tolerance [33], we made *a priori* predictions about the effect that some of these explanatory variables might have on age-specific CFR. For instance, we predicted that pathogens with broader tissue tropism (causing systemic infections) might cause more harm in older adults than pathogens that only infect one organ. This hypothesis is based on the rationale that aged hosts might be less able to

repair multi-organ damage, compared to repairing single organs (whatever the cause of the damage, immune- or pathogen-mediated), and therefore be more likely to succumb to the infection. Recent evidence based on human RNA viruses has indeed shown higher mortality following infection with viruses that induce systemic infections [37]. We predict that this relationship might be exacerbated in older adults. Following the same reasoning, we also predicted that pathogens that induce relatively long lasting disease symptoms should also be those associated with the highest CFR at old age, since older adults might be less prone to fully recover from long lasting symptoms compared to younger age classes. On the contrary, we do not have any particular prediction on how traits potentially related to human adaptation (i.e., human-to-human transmission, animal reservoir, emerging pathogens) or transmission mode might affect age-specific CFR.

## Materials and methods

### Data

We used data on age-specific mortality following infection with human pathogens that have been compiled by [20] and include 32 infectious diseases. These data are based on different mortality metrics and do not always report the number of infected cases and deaths. To allow a quantitative comparison, we therefore only used data for which numbers of infected people and number of deaths were reported. This restricted the dataset to 28 infectious diseases. For some of these infectious diseases, the data were derived from different countries or geographical regions and different periods, representing replicated datasets. In particular, for infectious diseases for which treatments and vaccines are available now, [20] sought older studies in order to have estimates of untreated CFR. Consequently, the data included spanned a large period, from the first half of the 19th century (cholera) to 2020 (COVID-19).

The data compiled by (20) reported heterogeneous age classes across different infectious diseases. Again, to allow a quantitative comparison, we clustered the number of infected and number of deaths using five-year intervals, from 0 to 80 years. This gave 16 age classes (ordered from 1 to 16). However, for some infectious diseases, the data covered larger age classes (e.g., 20 years) or referred to a minimum age (e.g., > 65 years). In this case, we averaged the corresponding age classes. For instance, when the data reported the number of infected and number of deaths for people between 25 and 45 years (corresponding to age classes 6, 7, 8 and 9 based on a 5 year interval), we attributed the data to the 7.5 age class (the average between 6, 7, 8 and 9). Similarly, when the data reported the number of infected and number of deaths for people older than 65 years, we attributed them to the age class 15 (the average between 14, 15 and 16). Sex was only reported for a small fraction of diseases and therefore it was not possible to systematically compare CFR between women and men across different ages. Accordingly, when reported, CFRs of women and men were combined in a unique value per age class. For COVID-19, we also reported sex-specific results.

To this initial dataset on age-dependent CFR, we added 11 variables referring to different clinical and ecological aspects of the disease: pathogen type (virus or bacterium), duration of the incubation period (three ordered classes), duration of symptoms/illness (three ordered classes), systemic/local infection (two classes), length of the human-pathogen association (two classes), animal reservoir (two classes), human-to-human transmission (two classes), transmission via ingestion (two classes), transmission via inhalation (two classes), transmission via contact with body fluids (two classes), transmission via vector (two classes).

All infectious diseases considered here are caused by either bacteria (Gram$^+$ and Gram$^-$) or viruses (DNA and RNA viruses). However, to avoid multiple categories with few diseases in some categories, pathogen type was scored using only two modalities (bacterial or viral

diseases). For incubation period and duration of symptoms/illness, values are often reported as ranges. In this case, we took the central value of the range and grouped them into three ordered classes ($<$ 1 week, 1 to 2 weeks, $>$ 2 weeks for incubation period, and $<$ 1 week, 1 to 3 weeks, $>$ 3 weeks for duration of symptoms/illness). For systemic/local infections, we used a 0/1 score where 0 referred to pathogens that produce local infections and 1 to pathogens that can spread systemically. For the length of the human-pathogen association, we used a 0/1 score where 0 refers to ancient and 1 to emerging pathogens. We considered as emerging, any new pathogen reported to induce a human disease from the 20[th] century onwards. Therefore, ancient diseases that have been re-emerging in the last decades (e.g., tuberculosis) were nevertheless scored as ancient. For the animal reservoir, we used a 0/1 score were 0 refers to pathogens that do not have any known animal reservoir from which humans can get the infection, and 1 refers to pathogens with known animal reservoirs. This definition, therefore, implies that pathogens that emerged from spillover events, but for which transmission is no longer sustained by animal reservoirs, were included in the 0 class (e.g., HIV, SARS-CoV-2). For the human-to-human transmission, we used a 0/1 score where 0 refers to pathogens for which humans represent dead ends, 1 refers to pathogens that can be transmitted from humans to humans (whatever the transmission mode). Finally, for transmission mode, we used four 0/1 scores: ingestion, inhalation, body fluids and vector. This allowed having multiple 1 for pathogens that can be transmitted by multiple pathways (e.g., Lassa fever, tuberculosis). S1 Table reports the list of diseases, the full description of the original information used to attribute scores for each variable, and all the sources the original information comes from. The complete dataset (the age-specific CFRs and all the explanatory variables used for the statistical models) is available in S1 Data.

## Statistical analyses

Age and infection-induced mortality can vary in a non-linear way. Therefore, we first used generalized additive models (GAMs) to assess the shape of the relationship between CFR and age for each infectious disease. GAMs decompose the linear and non-linear relationship between variables, using smoothing spline optimization functions. CFR was modelled as a binomial response variable (number of deaths/number of infected) and age was included as the independent variable. This allowed us to infer the shape of the relationship and to assess the significance of the linear and non-linear components. GAMs were run using PROC GAM in SAS (14.3).

Having characterized the shape of the relationship for each disease, we next implemented a finite mixture model with a beta-binomial distribution of errors (PROC FMM, SAS 14.3). We ran three sets of models that included different combinations of explanatory variables. We did this because including all the explanatory variables in the same model resulted in strong over-dispersion (even with a beta-binomial or a binomial cluster distribution). For each of these models, the response variable was the number of deaths/number of cases. The explanatory variables were included as follows, *model 1*: incubation period, duration of symptoms/illness, local/systemic infection; *model 2*: pathogen type, length of human-pathogen association, animal reservoir, human-to-human transmission; *model 3*: ingestion, inhalation, body fluids, vector. In addition to these explanatory variables, each model also included age (ordered class from 1 to 16), squared age, date (i.e., the period to which the CFR refers, ordered class), and geographical zone (whether the country/city the CFRs refer to lies in the intertropical zone, yes/no). Date and geographical zone were included to take into account a possible effect of temporal and spatial variation in CFRs. Finally, all models included the two-way interactions between age (and squared age) and the explanatory variables (incubation period, duration of

illness, local/systemic infection, pathogen type, length of association, animal reservoir, human-to-human transmission, ingestion, inhalation, body fluids, vector). Explanatory variables did not have strong associations (S1 Fig). Only i) pathogen type and length of human-pathogen association (all emerging pathogens being viruses), and ii) animal reservoir and vector transmission (all diseases that are not transmitted by vectors do not have animal reservoirs) were significantly associated.

Models were selected based on the Bayesian Information Criterion (BIC) [38]. We first ran an intercept only model and compared its BIC with models including age, squared age, date and intertropical zone. This was done to assess whether adding age (and squared age) improved model fit. We then added the explanatory variables as main effects and again assessed whether this improved model fit. Finally, we added the two-way interactions between age and the explanatory variables and between squared age and explanatory variables. Given that our aim was to investigate whether pathogen features account for age-specific CFR, we ran all the possible models with the squared age x explanatory variable interactions (in these models all the age x explanatory variable interactions were always included), and then all the possible models with the age x explanatory variable interactions (in these models all the squared age x explanatory variables had been removed). Therefore for *model 1*, we compared 19 models; for *model 2* and *3*, we compared 35 models each. Models with the lowest BIC values were considered as the most parsimonious ones. For each set of models, the ΔBIC (difference between the focal model and the model with the lowest BIC) was always higher than 2. Compared to the Akaike Information Criterion (AIC), BICs penalize complex models more strongly, and therefore provide more conservative results. S2, S3, S4, S6 Tables reports all the models run and the corresponding BICs, AICs, Pearson's statistics, and number of parameters. The SAS codes used to run GAMs and FMMs are also reported in S1 Text.

## Results

### Shape of the relationship between age and CFR

We could assess how CFR varies with age for 28 infectious diseases (Fig 1). GAMs showed that there was a significant linear trend for 26 of them (only for poliomyelitis and Lassa fever was there no statistical support for a linear component in the relationship between CFR and age) (Table 1). For the 26 infectious diseases with a significant linear trend, the sign of the relationship was positive for 23 of them, and only for dengue, pertussis, and diphtheria was the linear relationship between CFR and age negative (Table 1). Therefore, in 82% of the infectious diseases considered here, mortality increased with age, and only 11% tended to induce higher mortality in infants and lower mortality as age increases. In addition to the linear trend, GAMs also provided evidence for significant non-linear relationship between CFR and age in 26 out of 28 infectious diseases (Table 1, S2 Fig). AIDS was found to only have a linear relationship between CFR (% dying within 10 years post-infection) and age, while for Western equine encephalitis the statistical support for the non-linear trend was not significant after a sequential Bonferroni correction. The non-linear trend in most of the diseases was consistent with a slight decline during the first age classes and then a steady and often sharp increase in CFR with age.

For COVID-19, we also ran a model that included sex as a categorical explanatory variable. The model provided evidence suggesting that age-dependent mortality increased with a faster pace for men compared to women (S3 Fig and S7 Table).

### Clinical and ecological traits explaining age-specific CFR

We ran three sets of finite mixture models to assess the effect of several pathogen features (clinical and ecological traits) on age-specific CFRs.

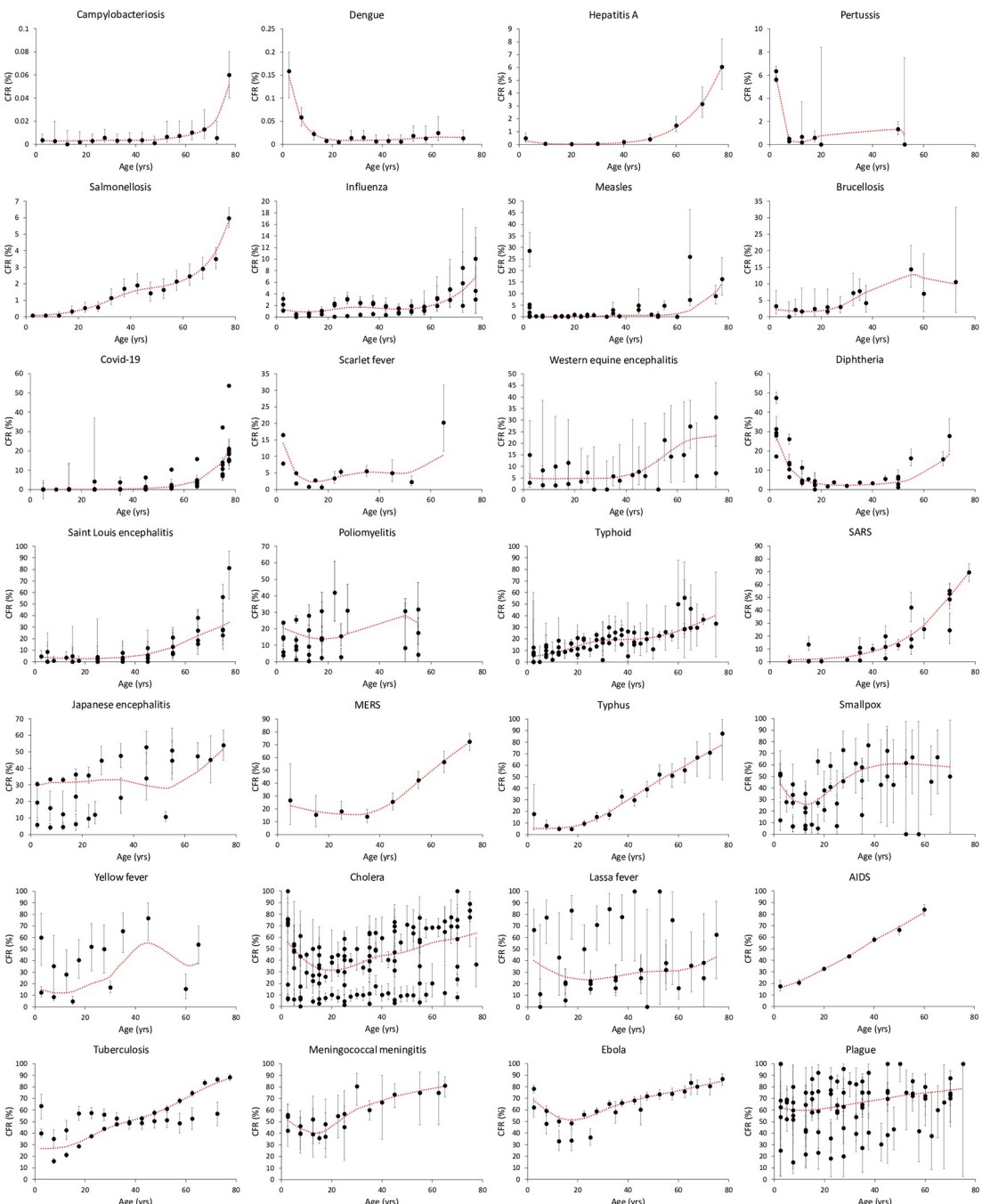

**Fig 1. Age-dependent variation of CFR (%) for the 28 human infectious diseases considered in the study.** Each dot represents the percent of infected people dying (number of deaths/number of cases) in each age class. Bars represent the binomial 95% confidence intervals. The dotted lines represent the fit of the GAMs. Multiple CFR values for a given age class refer to independent datasets.

*Model 1* included the incubation period, the duration of symptoms/illness and whether the pathogen produces a local or a systemic infection. Among the 19 competing models, the one with the lowest BIC value included the interaction between age and duration of symptoms/illness (Tables 2 and S2), and showed that CFR increased more with age for diseases producing long-lasting symptoms (Fig 2).

**Table 1. Generalized additive models (GAMs) investigating the shape of the relationship between CFR of 28 human infectious diseases and age.** For each model, we report the results for the linear (regression model analysis) and the non-linear (smoothing model analysis) components. The sign of the estimate indicates whether the linear trend was positive or negative. The models were run using a binomial distribution of errors (number of deaths/number of cases). Diseases are ordered by increasing values of CFR (deaths/cases). The column Replicates indicates the number of replicated datasets per disease. After a sequential Bonferroni correction, the non-linear trend for Western equine encephalitis was not statistically significant.

| Disease | Regression model analysis | | | Smoothing model analysis | | Deaths/Cases | Replicates |
|---|---|---|---|---|---|---|---|
| | *Estimate (SE)* | *t* | *p* | *$\chi$2* | *p* | | |
| Campylobacteriosis | 0.224 (0.024) | 9.25 | <0.0001 | 26.64 | <0.0001 | 81/1108979 | 1 |
| Dengue | -0.168 (0.020) | -8.21 | <0.0001 | 89.09 | <0.0001 | 154/804171 | 1 |
| Hepatitis A | 0.318 (0.018) | 17.93 | <0.0001 | 55.88 | <0.0001 | 138/45584 | 1 |
| Pertussis | -0.343 (0.021) | -16.55 | <0.0001 | 744.89 | <0.0001 | 3173/102453 | 2 |
| Salmonellosis | 0.236 (0.009) | 26.19 | <0.0001 | 56.28 | <0.0001 | 937/65099 | 1 |
| Influenza | 0.088 (0.008) | 10.96 | <0.0001 | 154.02 | <0.0001 | 966/68006 | 3 |
| Measles | 0.140 (0.005) | 25.51 | <0.0001 | 2614.60 | <0.0001 | 4041/2526156 | 7 |
| Brucellosis | 0.222 (0.034) | 6.46 | 0.0001 | 12.07 | 0.007 | 93/2395 | 2 |
| COVID-19 | 0.607 (0.005) | 130.11 | <0.0001 | 321.93 | <0.0001 | 23019/359799 | 7 |
| Scarlet fever | -0.344 (0.010) | -34.26 | <0.0001 | 1466.49 | <0.0001 | 6254/92106 | 2 |
| Western equine encephalitis | 0.145 (0.022) | 6.63 | <0.0001 | 8.05 | 0.045 | 99/1071 | 2 |
| Diphtheria | -0.354 (0.004) | -92.37 | <0.0001 | 5980.06 | <0.0001 | 35874/253245 | 5 |
| Saint Louis encephalitis | 0.233 (0.015) | 16.04 | <0.0001 | 26.38 | <0.0001 | 433/3580 | 4 |
| Poliomyelitis | 0.013 (0.009) | 1.35 | 0.189 | 82.82 | <0.0001 | 3078/16007 | 5 |
| Typhoid | 0.149 (0.005) | 28.04 | <0.0001 | 182.08 | <0.0001 | 4465/32116 | 8 |
| SARS | 0.399 (0.013) | 29.64 | <0.0001 | 12.69 | 0.005 | 822/5074 | 4 |
| Japanese encephalitis | 0.020 (0.006) | 3.37 | 0.0029 | 69.41 | <0.0001 | 7684/24688 | 3 |
| MERS | 0.253 (0.020) | 12.45 | 0.001 | 19.55 | 0.0002 | 408/1081 | 1 |
| Typhus | 0.332 (0.015) | 21.47 | <0.0001 | 15.39 | 0.002 | 704/3456 | 1 |
| Smallpox | 0.089 (0.014) | 6.42 | <0.0001 | 171.84 | <0.0001 | 1528/4091 | 6 |
| Yellow fever | 0.144 (0.021) | 7.01 | <0.0001 | 39.17 | <0.0001 | 232/1154 | 2 |
| Cholera | 0.033 (0.003) | 10.56 | <0.0001 | 1149.03 | <0.0001 | 12848/29306 | 14 |
| Lassa fever | 0.021 (0.019) | 1.10 | 0.282 | 15.33 | 0.002 | 344/1264 | 4 |
| AIDS | 0.261 (0.009) | 28.56 | 0.001 | 5.22 | 0.156 | 5420/12910 | 1 |
| Tuberculosis | 0.185 (0.003) | 54.61 | <0.0001 | 181.76 | <0.0001 | 19614/42730 | 2 |
| Meningococcal meningitis | 0.130 (0.017) | 7.69 | <0.0001 | 36.94 | <0.0001 | 832/1602 | 3 |
| Ebola | 0.082 (0.006) | 13.37 | <0.0001 | 236.26 | <0.0001 | 6416/10128 | 2 |
| Plague | 0.063 (0.008) | 7.75 | <0.0001 | 9.84 | 0.020 | 4220/6539 | 10 |

*Model 2* included a set of variables supposedly describing the potential of pathogens for human adaptation. Among the 35 competing models, the one with the lowest BIC included two two-way interactions: age x pathogen type and age x length of human-pathogen association (Tables 3 and S3). Age-dependent CFR increased more for bacterial diseases and for emerging pathogens (Table 3, Fig 3). However, it should be noted that among the 28 infectious diseases considered here, all the emerging ones involve viruses. We, therefore, tested whether emerging diseases exert a higher mortality burden in the oldest age classes using a restricted dataset with viruses only. Again, the model with the lowest BIC included the interaction between age and length of human-pathogen association, with emerging viruses producing the steepest increase in age-dependent CFR (Fig 3, S4 and S5 Tables).

Finally, *model 3* included four variables describing the mode of pathogen transmission. Among the 35 competing models, the one with the lowest BIC value only included the main terms (S6 Table), and all models with the interactions between age (and squared age) and the explanatory variables were rejected by the model selection procedure.

**Table 2. Finite mixture model with a beta-binomial distribution of errors exploring the effect of duration of symptoms, incubation period and whether pathogens induce a local or a systemic infection on age-specific CFR (number of deaths/number of cases).** The table reports the estimates (with SE and 95% CI), z and P values for the parameters retained in the model with the smallest BIC value. Number of observations = 873; number of deaths/number of cases = 143,877/5,624,790.

| Effects | Estimate (SE) | 95% CI | z | P |
|---|---|---|---|---|
| Intercept | 1.132 (0.259) | 0.624 / 1.641 | 4.37 | <0.0001 |
| Age | -0.123 (0.041) | -0.203 / -0.043 | - 3.02 | 0.0026 |
| Age$^2$ | 0.008 (0.002) | 0.003 / 0.012 | 3.56 | 0.0004 |
| Date | -0.493 (0.038) | -0.568 / -0.418 | -12.89 | <0.0001 |
| Intertropical (no) (yes) | -0.959 (0.102) 0 | -1.159 / -0.760 | -9.42 | <0.0001 |
| Duration of symptoms | -0.097 (0.092) | -0.278 / 0.084 | -1.05 | 0.2947 |
| Incubation period | 0.058 (0.066) | -0.071 / 0.187 | 0.88 | 0.3801 |
| Systemic infection (no) (yes) | -0.180 (0.098) 0 | -0.372 / 0.012 | -1.84 | 0.0658 |
| Age x duration of symptoms | 0.040 (0.011) | 0.019 / 0.061 | 3.79 | 0.0002 |

## Discussion

We assessed age-specific CFR for 28 major human infectious diseases and found that often, after a slight initial decrease, infection-induced mortality sharply increased with age. Although this pattern has already been described in previous reports [20], we went a step further and investigated whether some clinical and ecological traits of the infectious agents might account for differences in age-dependent CFR. In agreement with one of the predictions, we found that pathogens producing diseases with long-lasting symptoms are associated with the steepest increase in age-dependent CFR. We also found that bacterial infections and emerging viral diseases are associated with steepest increase in age-dependent CFR. Contrary to our prediction, we did not find evidence that pathogens producing systemic or local infections differ in their age-specific CFR. Similarly, the relationship between age and CFR did not differ among pathogen transmission modes.

Our work combines a hypothesis-driven and a descriptive approach. In other words, we had *a priori* predictions on how some of the clinical and ecological traits considered here should affect age-dependent CFR, but we also described the pattern of age-specific CFR according to pathogen traits for which we did not have an *a priori* prediction on the expected effect. Actually, we had two clear-cut predictions. First, we predicted that pathogens inducing a disease with long-lasting symptoms should be associated with steeper age-dependent mortality compared to diseases with faster resolving symptoms. Second, we predicted that infections

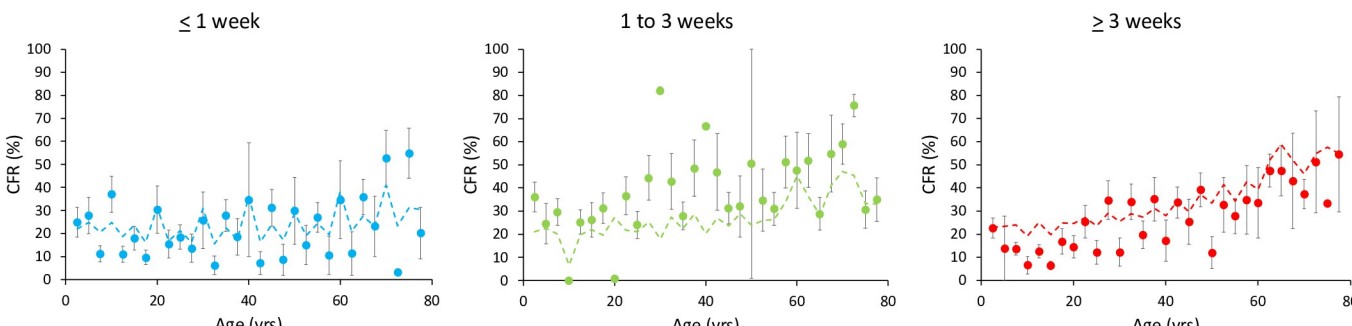

**Fig 2. Age-dependent variation in CFR (%) according to the duration of symptoms/illness.** Each dot represents the mean CFR for each age class and bars the standard errors. The dotted lines represent the fit of the finite mixture model with a beta-binomial distribution of errors (number of deaths/number of cases).

**Table 3. Finite mixture model with a beta-binomial distribution of errors exploring the effect of pathogen type, length of association, animal reservoir, and human-to-human transmission on age-specific CFR (number of deaths/number of cases).** The table reports the estimates (with SE and 95% CI), z and P values for the parameters retained in the model with the lowest BIC value. Number of observations = 873; number of deaths/number of cases = 143,877/5,624,790.

| Effects | Estimate (SE) | 95% CI | z | P |
|---|---|---|---|---|
| Intercept | 1.8312 (0.300) | 1.243 / 2.420 | 6.10 | <0.0001 |
| Age | -0.006 (0.040) | -0.085 / 0.074 | - 0.14 | 0.8894 |
| $Age^2$ | 0.006 (0.002) | 0.002 / 0.010 | 3.05 | 0.0023 |
| Date | -0.642 (0.042) | -0.725 / -0.560 | -15.26 | <0.0001 |
| Intertropical (no) (yes) | -0.709 (0.098) 0 | -0.901 / -0.516 | - 7.20 | <0.0001 |
| Pathogen type (bacterium) (virus) | 0.348 (157) 0 | 0.041 / 0.656 | 2.22 | 0.0265 |
| Length of association (ancient) (emerging) | -0.659 (0.228) 0 | -1.104 / -0.213 | - 2.90 | 0.0038 |
| Animal reservoir (no) (yes) | -0.332 (0.088) 0 | 0.020 / 0.061 | - 3.76 | 0.0002 |
| Human to human transmission (no) (yes) | -0.781 (0.153) 0 | -1.082 / -0.481 | - 5.09 | <0.0001 |
| Age x pathogen type (bacterium) (virus) | 0.077 (0.020) 0 | 0.038 / 0.116 | 3.86 | 0.0001 |
| Age x length of association (ancient) (emerging) | -0.099 (0.024) 0 | -0.146 / -0.053 | - 4.20 | <0.0001 |

that spread systemically should have more severe consequences in older adults compared to the younger age classes. Both predictions are based on the idea that age might compromise the capacity to tolerate the infection, in terms, for instance, of tissue repair [31]. Therefore, older adults might be less able to cope with damage caused over multiple organs (systemic diseases) and longer periods (diseases producing long-lasting illness). It should be fully acknowledged that infection-induced damage can arise both from a direct effect of pathogen multiplication and a poorly regulated inflammatory response [29]. Changes in immune functioning across ages point towards older adults being more likely to suffer from both types of damage compared to younger age classes. Indeed, depletion of naïve T cells, reduced responsiveness of B cells, or impaired phagocytic activity are among the many functional changes associated with aging which can account for reduction in host resistance (impaired capacity to limit pathogen proliferation) [24,25]. Concomitantly, older adults also experience major changes in their capacity to repair damaged tissues. In particular, a fibrotic response is often produced following tissue damage in older adults [24]. Fibrosis is characterized by a thickening of the damaged organ due to accumulation of connective tissue, resulting in a remodelling of tissue architecture and possibly organ failure. Fibrosis is orchestrated by immune pathways that are overexpressed in older adults (Th2 cytokines, TGF-β) and infection-induced fibrosis has been shown to occur more often in aged individuals [24]. For instance, long-term lung damage associated with SARS-CoV-2 is more likely to occur in older adults [39]. Our comparative analysis provides support for the hypothesis that infections inducing long-lasting symptoms are indeed relatively more lethal in older adults (but not systemic diseases) compared to diseases with rapidly resolving symptoms. Further work should attempt to better characterize whether this relationship stems from impaired resistance, tolerance or both in older adults.

In addition to these *a priori* predictions, we also explored whether age-specific CFR varies depending on some ecological traits of the pathogens that have been shown in previous reports to be associated with virulence. For instance, in a recent work, [11] suggested that facultative human pathogens might have lower virulence because they might be less well adapted to

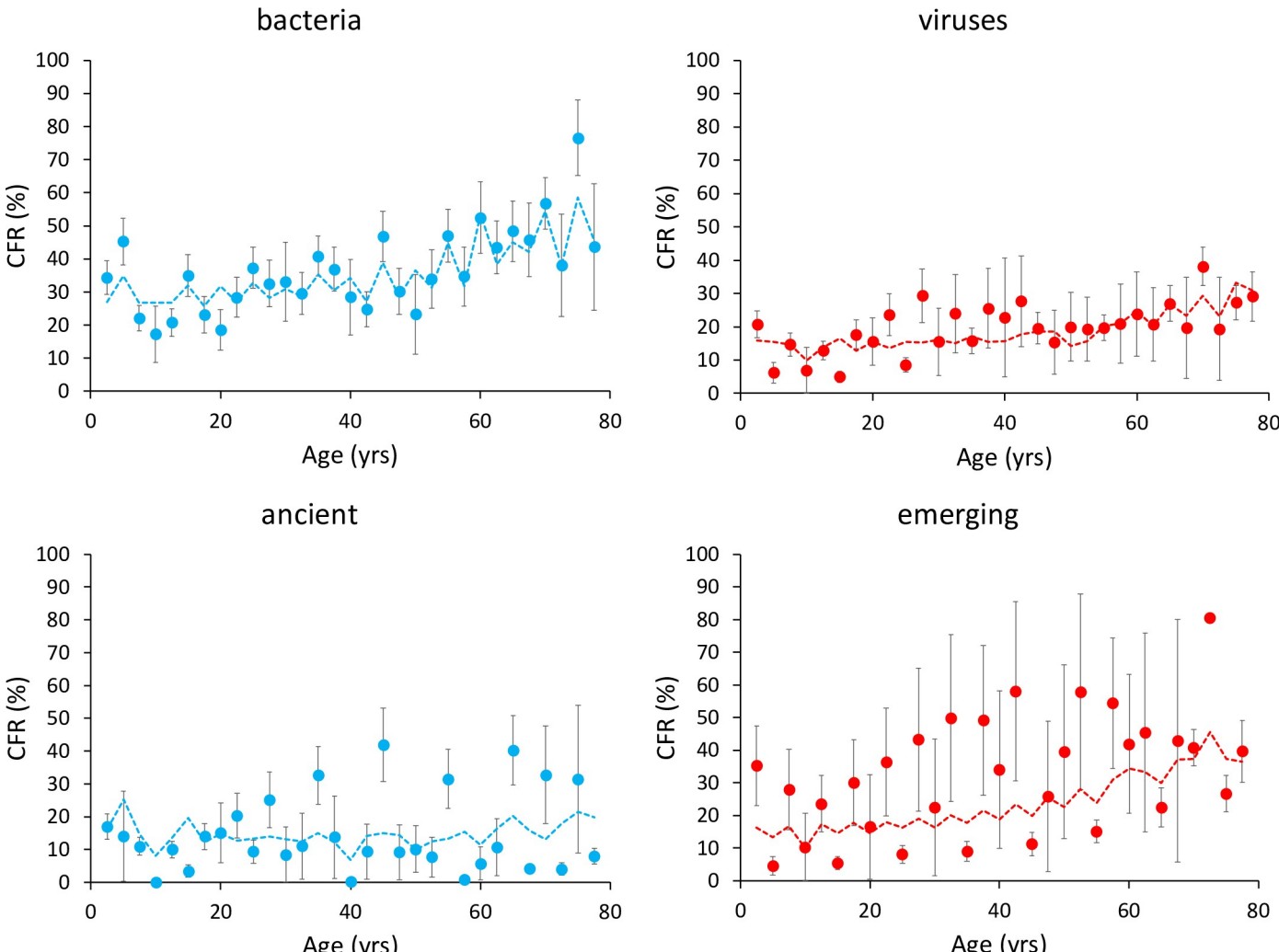

**Fig 3. Age-dependent variation in CFR (%) for bacterial and viral diseases (above) and emerging and ancient viral diseases (below).** Each dot represents the mean CFR for each age class and the bars the standard errors. The dotted lines represent the fit of the finite mixture model with a beta-binomial distribution of errors (number of deaths/number of cases).

humans. Here, we used a combination of ecological traits that should reflect the potential for pathogen adaptation to human hosts. These traits included whether humans are dead end hosts, whether pathogens have known animal reservoirs, and whether pathogens have recently emerged to infect humans. Overall, we did not find unequivocal evidence suggesting that age-dependent mortality might depend on the level of host adaptation. Among the three traits considered here, we found that emerging viruses are associated with steeper increase in age-dependent mortality, while neither human-to-human transmission nor animal reservoir were associated with how fast mortality increases with age. The reason why emerging viruses induce proportionally more damage in older adults are unclear, but again it is plausible to assume that the interaction between the pathogen and the immune response might play a role. More specifically, the pathology of several recently emerged viruses has been clearly identified as being due to hyper-inflammation and the associated damage (e.g., Ebola, Lassa virus, SARS-CoV-2) [40–42], which potentially makes aged patients more at risk for the reasons discussed above. However, this does not explain why emerging viruses might produce more exuberant (poorly

regulated) immune response than pathogens with a longer history of human infection. One might argue that immunopathology and the associated mortality reflect poor host adaptation. In agreement with this hypothesis, recent comparative evidence showed that infection-induced mortality of domestic animals increases when pathogens infect hosts that are outside of their documented host range [43]. However, other findings suggest that host shifts do not always produce an increase in virulence [44]. Finally, many spillover events might go unnoticed if they do not produce disease symptoms, skewing our perception of emerging diseases towards the most virulent pathogens.

Similarly, we did not find any evidence suggesting that the strength of the relationship between infection-induced mortality and age depends on the mode of pathogen transmission. Previous studies have suggested that transmission mode might shape parasite virulence, in particular vector-borne diseases might be associated with higher virulence because pathogens might still be transmitted by vectors even when the host is immobilized from the illness [45]. However, further work has questioned the relevance and generality of the prediction that pathogens transmitted by vectors should be more virulent (see [6] for a discussion). A recent comparative work has also shown increased virulence of human pathogens that are transmitted by inhalation [11], although the evolutionary reasons for this association are unclear. Although this was not our main aim, our analysis confirms that transmission mode does play a role on the virulence of human pathogens since we found that vector-borne diseases and pathogens that are transmitted through contact with body fluids have on average higher CFRs (S8 and S9 Tables; S4 Fig). However, again, older people do not appear to be proportionally more vulnerable neither to vector-borne diseases nor to pathogens transmitted through contact with body fluids.

Finally, we found that the relationship between age and CFR was steeper for bacterial than for viral diseases. This pattern appears to be driven by the high mortality burden associated with bacterial infections in the pre-antibiotics era (as mentioned above [20] sought old reports of infection-induced mortality to avoid the confounding effect of drugs on CFR). The evolution of antibiotic resistance currently poses a major threat to human health at the global scale [46]; in this context, our analysis suggests that older adults might be particularly at risk [47–49].

Pathogen fitness is thought to be governed by trade-offs between within-host replication and between-host transmission [6,50]. On one hand, pathogens with high within-host replication might be more successful at escaping immune clearance (which increases the duration of the infection) and might have higher chance to transmit to other hosts; on the other hand, pathogens with high within-host replication might also be more likely to kill the host which obviously reduces the duration of the infection and precludes further transmission to other hosts. Therefore, according to this trade-off model, pathogen fitness is maximized at intermediate levels of virulence (i.e., infection-induced mortality) [51]. However, when host background mortality (infection-independent mortality) increases with age, the strength of the selection on virulence is also likely to vary [6]. Therefore, since background mortality at young ages is low, pathogens might be rewarded by adopting a prudent strategy of host exploitation resulting in prolonged opportunities for transmission [52]. In aged hosts, where background mortality is higher, selection should favour more aggressive pathogen strains [52]. In this case, the finding of increased infection-induced mortality as a function of host age might also result from a pathogen strategy to increase host exploitation when opportunities for transmission are reduced. This corresponds to a shift from a host-centred view of infection-induced age-dependent mortality (e.g., due to impaired immunity at old age) to a pathogen-centred view where selection acts differently on pathogens depending on host age. This approach has been recently adopted to explain the evolution of sex-specific virulence [53]. Experimental evolution of

pathogen virulence in response to host age has been reported in coxsackievirus B3 (CVB3) [54]. Avirulent CVB3 infecting old mice reach higher titres (viremia) compared to infection of young hosts. Interestingly, after passaging in old mice, avirulent CVB3 acquire a profile of increased virulence (higher viremia and pathology) even when infecting young hosts. This shows, in agreement with the theoretical argument, that host age promotes pathogen virulence. Such host age driven evolution should be more likely to occur when hosts tend to cluster in age-homogeneous groups where pathogens are transmitted within same-age hosts. Clusters of age-homogenous hosts are frequent in humans at both extremes of the age distribution (nurseries, kindergartens, schools at one end, and retirement communities, nursing homes, etc. at the other end).

## Limitations

Although we restricted our modelling effort only to diseases for which number of cases and number of deaths were reported [20], some heterogeneity does persist in the data. For instance, antibiotics and vaccination programs have substantially reduced the overall mortality risk due to infection [3,46]. In order to reduce the bias due to vaccine and drug administration, [20] used old reports of the 19th century for ancient diseases, prior to antibiotic discovery and widespread vaccination programs. While this indeed provides values of untreated CFR, it nevertheless might introduce another bias related to the accuracy of the diagnostic, the actual cause of mortality, or the overall health conditions that obviously have changed over the last 150 years [3]. We attempted to correct the heterogeneity due to the large period covered by the data (from 1832 to 2020), by including this variable in all statistical models, which allowed us to infer the effect of the other variables independently from the period when the data were gathered.

Another difficulty inherent to this type of comparative analyses comes from the necessity to boil down information that usually comes in a qualitative form into quantitative variables that can be fed into statistical models. This applies to information related to the clinical traits included in our study such as incubation period and duration of symptoms/illness, for which values are usually reported as ranges. We used central values when ranges were reported but these central values should be considered with caution because the variance around them can be considerable. For this reason, we clustered the values into relatively large classes which therefore should buffer the variation around these central values. However, we acknowledge the uncertainty associated with these values.

We would also like to stress that, contrary to CFR, the information on clinical traits are not age-specific. In other terms, while it is reported that, for dengue, incubation time ranges from 4 to 10 days, we do not know if incubation time itself depends on host age. This might even be more relevant for duration of symptoms/illness for which it seems plausible to expect that young individuals might have faster resolving symptoms than older adults. Further work should attempt to include age-specific information on clinical variables.

To conclude, our analysis shows that age is a key trait affecting infection-induced mortality rate in humans, with the rate of increase in mortality with age largely exceeded the initial decline. Therefore, age is an intrinsic host factor that consistently aggravates the outcome of the infection, and the aggravating effect depends on both clinical and ecological variables of the pathogen.

## Supporting information

**S1 Table. List of human infectious diseases used to investigate age-specific case fatality rate (CFR).** For each disease, we report the pathogen, the pathogen type, the pathophysiology,

the incubation period, the duration of symptoms/illness, the transmission route, the length of the human pathogen association, the clinical symptoms, whether the pathogen is known to have an animal reservoir, whether the pathogen can be transmitted from human to human. This information was used to attribute the values for each of the variables included in the finite mixture models. Sources from which this information was extracted are given at the bottom of the table.
(DOCX)

**S2 Table. Model comparison on the effect of duration of symptoms/illness, incubation period, local/systemic infection on age specific CFR for 28 human infectious diseases.** We report the -2 Log Likelihood, AIC, BIC, Pearson Statistics, number of parameters (k), the overdispersion parameter (Pearson Statistic/(N-k), and the ΔBIC. N = 873 observations. We ran 19 competitive finite mixture models. The model with the lowest BIC value is highlighted in green.
(DOCX)

**S3 Table. Model comparison on the effect of pathogen type, length of human-pathogen association, animal reservoir and human-to-human transmission on age specific CFR for 28 human infectious diseases.** We report the -2 Log Likelihood, AIC, BIC, Pearson Statistics, number of parameters (k), the overdispersion parameter (Pearson Statistic/(N-k), and the ΔBIC. N = 873 observations. We ran 35 competitive finite mixture models. The model with the lowest BIC value is highlighted in green.
(DOCX)

**S4 Table. Model comparison on the effect of length of human-pathogen association, animal reservoir and human-to-human transmission on age specific CFR for the restricted dataset including only viral diseases.** We report the -2 Log Likelihood, AIC, BIC, Pearson Statistics, number of parameters (k), the overdispersion parameter (Pearson Statistic/(N-k), and the ΔBIC. N = 443 observations. We ran 19 competitive finite mixture models. The model with the lowest BIC value is highlighted in green.
(DOCX)

**S5 Table. Finite mixture model with a beta-binomial distribution of errors exploring the effect of length of human-pathogen association, animal reservoir, and human-to-human transmission on age-specific CFR (number of deaths/number of cases) for the restricted dataset including only viral diseases.** The table reports the estimates (with SE and 95% CI), z and P values for the parameters retained in the model with the lowest BIC values. Number of observations = 443; number of deaths/ number of cases = 54,782/3,884764.
(DOCX)

**S6 Table. Model comparison on the effect of transmission by body fluids, ingestion, inhalation, vectors on age specific CFR for 28 human infectious diseases.** We report the -2 Log Likelihood, AIC, BIC, Pearson Statistics, number of parameters (k), the overdispersion parameter (Pearson Statistic/(N-k), and the ΔBIC. N = 873 observations. We ran 35 competitive finite mixture models. The model with the lowest BIC value is highlighted in green.
(DOCX)

**S7 Table. Generalized additive model (GAM) investigating the shape of the relationship between COVID-19 CFR and age in women and men.** We report the results for the linear (regression model analysis) and the non-linear (smoothing model analysis) components. The model was run using a binomial distribution of errors (number of deaths/number of cases).
(DOCX)

**S8 Table. Model comparison on the effect of transmission by body fluids, ingestion, inhalation, vectors on age specific CFR for 28 human infectious diseases.** We report the -2 Log Likelihood, AIC, BIC, Pearson Statistics, number of parameters (k), the overdispersion parameter (Pearson Statistic/(N-k)), and the ΔBIC. N = 873 observations. We ran 19 competitive finite mixture models to identify the model with the lowest BIC. Only models with the main effects (no interaction) were compared here. The model with the lowest BIC value is highlighted in green.
(DOCX)

**S9 Table. Finite mixture model with a beta-binomial distribution of errors exploring the effect of transmission mode (contact with body fluids, ingestion, inhalation, vector) on age-specific number of deaths/number of cases.** The table reports the estimates (with SE and 95% CI), z and P values for the parameters retained in the model with the lowest BIC value. Number of observations = 873; number of deaths/ number of cases = 143877/5624790.
(DOCX)

**S1 Fig. Heatmap of the association (contingency coefficients) between explanatory variables considered here.** The redder the color the stronger the association. A Fisher's exact test with a sequential Bonferroni correction showed that only the associations between pathogen type and length of human-pathogen association (p = 0.0084) and between animal reservoir and vector (p = 0.0058) were statistically significant.
(DOCX)

**S2 Fig. Smoothing component analysis of the effect of age on CFR (number of deaths / number of cases) for the 28 human infectious diseases considered here.** Each panel shows the spline fit (with the 95% CI) of generalized additive models run for each disease, after the linear trend has been removed. For instance, the fact that the spline fit for the relationship between CFR and age for AIDS is a flat line with 95% CI overlapping zero over the whole range of ages indicates that CFR only varies linearly with age. On the contrary, the spline fit for cholera shows a strong non-linear pattern.
(DOCX)

**S3 Fig. Age-dependent variation of CFR (%) for COVID-19 in women (red dots and line) and men (blue dots and line).** Each dot represents the percent of infected people dying (number of deaths/number of cases) in each age class. Bars represent the binomial 95% confidence intervals. The dotted lines represent the fit of the GAM. Multiple CFR values for a given age class refer to independent datasets.
(DOCX)

**S4 Fig.** CFR (%) for diseases transmitted through contact with body fluids (A), and diseases transmitted through vector bites (B). We report the mean ± SE.
(DOCX)

**S1 Text. SAS codes.**
(DOCX)

**S1 Data. Dataset.**
(XLSX)

## Author Contributions

**Conceptualization:** Gabriele Sorci, Bruno Faivre.

**Data curation:** Gabriele Sorci, Bruno Faivre.

**Formal analysis:** Gabriele Sorci.

**Funding acquisition:** Gabriele Sorci.

**Investigation:** Gabriele Sorci, Bruno Faivre.

**Visualization:** Gabriele Sorci.

**Writing – original draft:** Gabriele Sorci.

**Writing – review & editing:** Gabriele Sorci, Bruno Faivre.

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
