## [Decision Letter · Decision Letter 0]

16 Aug 2022

Dear Dr. Sorci,

Thank you very much for submitting your manuscript "Age-dependent virulence of human pathogens" for consideration at PLOS Pathogens. As with all papers reviewed by the journal, your manuscript was reviewed by members of the editorial board and by several independent reviewers. The reviewers appreciated the attention to an important topic. Based on the reviews, we are likely to accept this manuscript for publication, providing that you modify the manuscript according to the review recommendations.

Sincerely,

Raul Andino

Section Editor

PLOS Pathogens

Raul Andino

Section Editor

PLOS Pathogens

Kasturi Haldar

Editor-in-Chief

PLOS Pathogens

orcid.org/0000-0001-5065-158X

Michael Malim

Editor-in-Chief

PLOS Pathogens

orcid.org/0000-0002-7699-2064

Reviewer Comments (if any, and for reference):

Reviewer's Responses to Questions

**Part I - Summary**

Reviewer #1: The strength of this study is the detailed examination of the age-related impact of multiple diseases. This will advance the important conversation about tolerance and resistance in general and will inform research into mechanisms.

Reviewer #2: This work leverages a very large dataset (compiled and previously published by other authors in 2020) to ask how case fatality rates (CFRs) vary with age in humans, across a wide range of bacterial and viral infections. This is an important question, of general interest to anyone interested in any aspect of infection, immunity and public health.

At first, I was a little confused by this paper, because the publication from where the data is repurposed here, also presents age-dependent CFRs for many infectious disease, and also concludes that some age classes are more susceptible to a range of infection. So there was immediately a question of novelty (or lack thereof) and I was also left feeling a little uncomfortable that the authors who originally complied and analysed the data and addressed a similar question were not co-authors.

It later became clearer, that the authors' aim were to stratify the current data set by a number of further features of each pathogen: "For each infectious disease included in the dataset, we added the following information: 1)

whether the infectious agent is a virus or a bacterium; 2) the duration of the incubation period;

3) the duration of the disease symptoms; 4) whether the infection is localized or systemic; 5)

whether the infectious agent can be transmitted from human-to-human; 6) whether the

infectious agent has an animal reservoir; 8) whether the infection agent has a recent history of

human infection (i.e., has emerged in 20th century); 7) whether the infectious agent is

transmitted by ingestion, inhalation, contact with body fluids, or by vectors.

This is indeed a valuable addition to the previous analysis, and of great interest to the community. So this is in fact quite novel and also of great significance to our understanding of disease progression.

The data appear very well curated and the analyses seem sound, using Generalised Additive Models (GAMs) to incorporate both linear and non-linear components of the relationship between CFR and age.

The results are generally clear: with some notable excpetions (Dengue, Campylobacter) the CFR increases with age in a linear way with age. Other insight include:

that pathogens producing diseases with long-lasting symptoms are associated with the steepest increase in age-dependent CFR.

bacterial infections and emerging viral diseases are associated with steepest increase in age-dependent CFR.

No evidence that pathogens producing systemic or local infections differ in their age-specific CFR, or that the relationship between age and CFR did not differ among pathogen transmission modes

**Part II – Major Issues: Key Experiments Required for Acceptance**

Reviewer #1: Not applicable concerning experiments, but they should present results by sex for Covid-19.

Reviewer #2: I have no major comments regarding improvement to the current analyses and results, but some patterns were not really explained properly. For example, the decline in DFR with age for dengue, pertussis, and diphtheria is never really discusses or explained.

Another pertinent point is that by definition, CFR can refer to probability of dying after contracting the infection, in the absence or presence of pharmaceutical treatments. In this case, many of these disease analysed have access to numerous treatments, both prophylactic and curative. These are likely to affect the CFR in an age-specific way (and indeed the treatments themselves may have an age-specific pattern of distribution and efficacy). It isn't clear how this was dealt with, if at all, and whether the CFR analysed here is in the presence or absence of infection, or a mixture of both.

**Part III – Minor Issues: Editorial and Data Presentation Modifications**

Reviewer #1: Line editing:

27 whether pathogen characteristic might explain

41 fatal outcome

43 pathogens interact

50 in older adults

57 Throughout

60 has been dashed

63 often conceived

90 immune function

108 in the proper regulation of

118 After they enter the host

126 it has been suggested that this could affect virulence

131 has already been

133 Here we try to fill... To this end, we used

143 made a priori predictions about

144 we predicted [don't be tentative] ... with broader tissue tropism

169 the data included

173 [on the x axis of the figure, report the midpoints of the ages, not the numbers of the age classes]

183 [It might be worth reporting the CFRs by age for males and females separately for Covid]

206 pathogens that emerged

224 allowed us to

230 We did this because

237 to which the CFR refers, [delete the second "to"]

244 Explanatory variables did not have strong associations

263 Compared to the Akaike

273 was there no

275 diphtheria was the ... age negative.

286 [outweighs in what sense? What you are trying to say here is not clear]

327 we also described

344 also experience major

351 provides support for the

354 whether this relationship

368 are unclear, but

374 responses than pathogens

375 reflect poor host

379 host shifts do not

390 for this association

391 was not our main aim, our

430 kindergartens [English uses the German word]

435 [how this effect works is not clear from your compressed explanation. please expand it/]

433 reported (20), some heterogeneity

448 another bias

461 However, we acknowledge the uncertainty

470 To conclude, our analysis shows

Reviewer #2: Minor comment:

I feel the authors could more explicit about what the previous did NOT analyse. At several points the authors are upfront that this is a reanalysis if published data, but it took a while to finally realise what this study has added to the previous one. A sentence or two in the introduction explicitly stating what was not investigated would be helpful.

PLOS authors have the option to publish the peer review history of their article (what does this mean?). If published, this will include your full peer review and any attached files.

Reviewer #1: **Yes: **Stephen C. Stearns

Reviewer #2: No

Figure Files:

Data Requirements:

Reproducibility:

References:

---

## [Editor Report · Decision Letter 1]

8 Sep 2022

Dear Dr. Sorci,

We are pleased to inform you that your manuscript 'Age-dependent virulence of human pathogens' has been provisionally accepted for publication in PLOS Pathogens.

Best regards,

Raul Andino

Section Editor

PLOS Pathogens

Raul Andino

Section Editor

PLOS Pathogens

Kasturi Haldar

Editor-in-Chief

PLOS Pathogens

orcid.org/0000-0001-5065-158X

Michael Malim

Editor-in-Chief

PLOS Pathogens

orcid.org/0000-0002-7699-2064
---

## [Editor Report · Acceptance letter]

14 Sep 2022

Dear Dr. Sorci,

We are delighted to inform you that your manuscript, "Age-dependent virulence of human pathogens," has been formally accepted for publication in PLOS Pathogens.

Best regards,

Kasturi Haldar

Editor-in-Chief

PLOS Pathogens

orcid.org/0000-0001-5065-158X

Michael Malim

Editor-in-Chief

PLOS Pathogens

orcid.org/0000-0002-7699-2064